# The Active Role of Job Crafting in Promoting Well-Being and Employability: An Empirical Investigation

Fulvio Signore [1,*], Enrico Ciavolino [2,3], Claudio Giovanni Cortese [4], Elisa De Carlo [2] and Emanuela Ingusci [2]

1 Department of Humanities, University of Foggia, 71122 Foggia, Italy
2 Department of Human and Social Sciences, University of Salento, 73100 Lecce, Italy; enrico.ciavolino@unisalento.it (E.C.); elisa.decarlo@unisalento.it (E.D.C.); emanuela.ingusci@unisalento.it (E.I.)
3 Department of Computer Science and New Technology, WSB University, 80-266 Gdańsk, Poland
4 Department of Psychology, University of Turin, 10124 Turin, Italy; claudio.cortese@unito.it
* Correspondence: fulvio.signore@unifg.it

**Abstract:** Background: Job crafting is a proactive behavior displayed by workers to modify the boundaries of their roles, adapting them to their own needs, which is positively associated with motivational processes and negatively associated with mechanisms that compromise well-being. Methods: Starting from this framework, the objective of this study is to assess the strategic role of job crafting in the relationship between job insecurity and work engagement, psychological well-being and emotional exhaustion, and also to specifically investigate how much age impacts these behaviors and the relationship between job crafting and employability. The hypotheses were explored using partial least squares structural equation modeling (PLS-SEM) and validated using 5000 bootstrap re-samples, differentiating the models by the type of contract and organization of origin. Results: The results confirm the crucial role of job crafting in improving individual well-being and increasing processes of higher expendability in the job market and its protective role against stress mechanisms. Conclusions: Therefore, the results highlight the potentially beneficial effects of job crafting interventions, which are capable of providing tools to facilitate individual and cultural growth.

**Keywords:** job crafting; JD-R; PLS-SEM; multi-group; well-being; stress; contract; sector; sustainability

## 1. Job Crafting as a Sustainable Tool in Turbulent Times

Over the past 25 years, the contemporary professional landscape has changed radically. This change was the complex result of sequences of emerging economic, organizational, and social trends [1]). For this reason, both organizations and workers have undergone substantial changes. Both are required to be more flexible in adapting to various changes in order to respond more quickly and efficiently to market changes. Several studies have highlighted the importance of enhancing job resources to increase employability through skills [2] and to support the development of a sustainable career [3]. These studies emphasized, in particular, the importance of personal resources and flexible employment in reducing work demands and improving well-being [4]. Moreover, an increasing accent on the meaningfulness of work and person–job fit has resulted in an ever more central role for subjective career success as an indicator of successful career development [5].

This emphasizes, on the one hand, the role of employability, and on the other hand, the role of proactive behavior in increasing well-being and perceived career success [2].

In fact, since 2000, the topic of job crafting has become crucial and strategic for organizations, especially in terms of sustainability. As a matter of fact, it is true that the economic and professional context in the post-financial and health crisis has led to enormous structural changes; consequently, many companies are looking for sustainable competencies, which can allow the company to survive, even in emergency situations. Job

crafting, a pro-active behavior, was studied in empirical terms as an antecedent, mediator, and moderator of positive and negative outcomes [6–10]. The contextualization of job crafting is split into two broad research strands: qualitative and quantitative. The former, based on Wrzesniewski and Dutton's definition [11], defines job crafting as a process used by employees to "mould" their work in terms of physical and cognitive interventions which modify work boundaries on a cognitive level in terms of tasks or relationships [12]. The latter, which makes sense within the job demands–resources model [13], quantitatively defines job crafting in view of the changes that employees can implement to reconcile their demands and professional resources with their abilities and personal needs [12]. From this perspective, job crafting is operationalized as an attempt to increase structural resources—that is, competencies, abilities, and opportunities for professional development; to increase social resources—that is, relationships, feedback, and support of managers and colleagues; to increase challenging demands—that is, those that favor personal growth and learning; and to reduce disruptive demands—that is, circumstances which are emotionally impactful on health and individual well-being [14].

Some empirical research has made it possible to identify a positive association between job crafting and an increase in motivational processes and, at the same time, a negative association regarding outcomes connected to compromising one's health. As a matter of fact, different studies have highlighted how good job crafting practices can diminish the effects of burnout (or exhaustion) or stress in general [15–17] and, at the same time, promote behaviors with higher work engagement connected to improving the motivational process [18]. As hypothesized by [19], job crafting proves to be a powerful personal tool to foster the sustainability aspects of organizations, such as their ability to innovate. According to this study, in fact, job crafting has a positive impact on organizational innovation, defined as "*transformation in the organization, such as organizational structure, procedures, administrative systems, performance knowledge management, and managerial skills that can motivate them to use resources more successfully*" by [20], with a clear reference to its sustainability [21]. Furthermore, as proposed [22], the positive dimensions of job crafting seem to activate employability resources in workers from a reciprocal relationship perspective.

Within this framework, this study is an assessment of the beneficial role of job crafting in a framework aimed at improving the business conditions, quality of organizational life, and characteristics that make organizations healthy environments for employees and productivity [23,24].

More specifically, this study aims to explore the effect of job crafting on the relationship between job insecurity and work engagement, psychological well-being, and emotional exhaustion, exploring the role of age in creating proactive behaviors. In fact, according to different studies [15,25], age has a direct influence on job crafting behavior, so this study is aimed, from the perspective of deepening sustainable competencies, at verifying these assumptions. In addition, the investigation intends to verify whether job crafting can lead to spontaneous actions on behalf of workers aimed at higher employability. Although there are numerous studies that involve multiple variables in measuring well-being in different types of workers [26,27], and above all from a sustainability perspective, this research intends to measure how a construct related to proactivity, and therefore part of personal sustainability [28], can activate behavior that, according to the JD-R model, is categorized as positive and negative outcomes by taking some of them as examples. In addition, the study is aimed at how such behavior does not depend on conditions unfortunately made more frequent by the pandemic, such as job insecurity [29,30].

In this framework, a multi-group analysis was carried out to differentiate the models by type of contract (temporary/permanent) and organization of origin (public/private) to assess any differences in the model based on the recommended classification factors. The hypothesis, formulated based on the JD-R theoretical model, is explained by considering the distinction proposed by Bakker and Demerouti [13,31,32] between professional demands, professional resources, and outcomes correlated with them.

## 2. The Theoretical Framework: The Job Demands–Resources Model

The job demands–resources model (JD-R; [31,33,34]) was created as an operational theory aimed at explaining the etiology of stressful processes. Due to its simplicity, it appears to be one of the most flexible and usable models in professional environments and beyond. According to this model, every professional context can be read according to two specific dimensions: job demands and resources. More specifically, demands encompass all those physical, organizational, social, and psychological aspects that characterize a job and require effort and psycho-physical expenditure on behalf of the worker. Resources include all the physical, organizational, social, and psychological aspects, which allow the worker to achieve their work goals, minimize the psycho-physical costs required to meet professional demands, and encourage personal growth [34].

Bakker et al. [31] specify that demands are not only necessarily negative work aspects: as a matter of fact, there are also professional dynamics that could be categorized into demands which, however, act as an incentive for workers, in the sense that they promote professional and personal growth processes. Similarly, resources are not useful simply to manage demands; they also have positive values themselves. Resources can be divided into personal resources [35] and organizational resources. The former encompasses, for example, competencies, self-efficacy, and psychological capital [35].

Demands and resources, at a time when it is not possible to correctly balance the two, generate two independent psychological processes: the excessive presence of demands, such as a heavy workload, role ambiguity, and little control, can compromise one's health, which in the long term can lead to different stress-correlated outcomes (depression, burnout, physical illnesses, etc.). On the contrary, the presence of resources such as autonomy, feedback, and support on behalf of colleagues can activate a motivational process, which promotes the involvement and willingness to learn and achieve better performances.

### 2.1. The Study Included Variables

### 2.1.1. Job Insecurity

One's work environment is often characterized by objective and subjective insecurity, which is perceived by the individual based on an involuntary experience that generates feelings of helplessness and lack of control. These considerations lead to the creation of the concept of job insecurity, defined as a condition of great uncertainty about one's employment status, which can lead to time-based pressure, interpersonal conflicts, great concern about the future, the anticipation of "job loss" [36], and many other negative consequences pertaining to the workers' well-being [37]. The feelings generated by the conditions of job insecurity can be chronic and cause a great deal of stress, which is why reflecting on how these perceptions are managed is necessary. The construct of work insecurity is generally conceptualized in two ways: in a uni-dimensional way and in a multi-dimensional way; whereas, the former focuses on the mere perception of job loss. The latter illustrates that—in addition to the fear of losing work—there are elements which are strongly linked to one another, such as fear of loss of certain work conditions, position in the company, career opportunities, career growth, job description, and salary [38]. The first attempts at a multidimensional conceptualization of such a construct trace back to Greenhalgh and Rosenblatt [39], who stated that a uni-dimensional measure cannot adequately reflect the multi-faceted reality of job uncertainty; therefore, they suggested including changes to the work characteristics in the definition of its concept, together with a sense of powerlessness when combating job threats and their characteristics [40]. In an empirical framework, different studies agree that job uncertainty is subjective and, as such, can be interpreted in different ways by different workers [41]. As a matter of fact, if there are workers who experience panic and anxiety, there are others who do not ascribe value to professional continuity or even take it into consideration [42]. Some variables have proven to be suitable for managing and minimizing the consequences of job insecurity [43], including individual and demographic factors and proactive behavior. Among personal factors, a key role is played by internal control (high levels lead the individual to experience

fewer negative reactions towards insecurity) and psychological capital [41]. Among the demographic factors, age has the most negative effect on people's health. The threat of unemployment has a greater influence on older workers as it represents a threat to their very identity, especially when this threat occurs closer to their retirement.

On the contrary, younger workers display lower implications in terms of identity loss [41]. Another aspect that impacts the perception of job insecurity is job seniority: being part of an organization for a very long time leads to higher job involvement [44], which can have an impact on the uncertainty of keeping one's job. Variables such as age, gender, type of education, salary, type of contract, and type of work have a noticeable impact on the perception of insecurity. Proactive behavior, such as employability and job crafting, can reduce the consequences of job insecurity. Lastly, the individual's perception when it comes to taking on future jobs and the chance of finding another job [45] affect them, making insecurity less impactful. Although job crafting seems to play an indirect role in job insecurity, a study by Oprea and Ilescu [46] shows how the relationship between burnout and job insecurity is mediated by the desire to be challenged, which is a key aspect of job crafting.

### 2.1.2. Employability

The growing dynamism of the job market requires individuals to be more flexible and adaptable to the requirements of given circumstances. The new perspectives in terms of career paths highlight how important it is for workers, even if already employed, to raise their level of "attractiveness" in the professional sphere [47]. With "employable" or "employability", we refer to the ability to obtain and maintain a job over time and to manage any job transition which could manifest in the event of an organization being acquired by another or in the event of a change in roles; it is the ability to adapt to the changing requests of the external context [48]. Employability is a "multidimensional work-specific form of active adaptation, which enables the worker to identify and create new career opportunities" [49]: it is a strategy of continuous upgrading of one's competencies and abilities to keep up with changes in the professional world and to progress in one's career. An adaptable individual can change a series of personal elements (knowledge, abilities, dispositions, and behaviors) to meet the needs of the circumstances they are in [50]. Employability cannot exclusively depend on what the employer wants from their employees; it must also hinge on what an individual commits to doing to increase their internal or external career success [51,52]. In the literature, different authors see employability as a construct built around the individual. This construct helps to understand how people can promote better levels of adaptation to face the changes in the current working environment [49]. According to Lo Presti and Pluviano and Lo Presti et al. [51,52], employability is a dynamic construct, and it can be identified through four dimensions which are interconnected: these are career identity; management of the self and professional development; the creation of social networks; and environment monitoring. The theoretical model proposed by the authors confirms that employability is a resource which can comprehend and improve individual experiences in the job market. Employability is a central variable as it represents an individual strategy towards the continuous upgrading of competencies relevant for re-employment. Different studies have explored the above-mentioned variables and the consequences of employability. In general, among the antecedents, there are situational factors such as the demand/supply of work or individual factors, e.g., human and social capital, dispositions, or personality traits. Other studies included self-efficacy and job research behaviors among the variables impacted by employability.

### 2.1.3. Work Engagement

The recent conceptualizations and theoretical models developed in work psychology have allowed for the expansion of the interest of researchers and scholars in general; they are now not only interested in processes that compromise health as a result of work conditions, but also in a complementary mechanism, as hypothesized by the job demands–resources

model, or the one which implements motivational processes [53,54]. The recent positive psychological perspective highlights "the study and application of the key points and psychological abilities and human resources which may be measured, developed and efficiently managed for the improvement of performances in the workplace today" [55] (p. 59) in contrast with the pre-eminent pathologizing conception. One of the best-explored constructs in this field is undoubtedly work engagement. Work engagement is described as a "positive, gratifying state of mind, connected to work and characterized by vigor, dedication and absorption" [56], (p. 74). Therefore, the definition foresees the existence of three sub-dimensions which are connected to one another: vigor refers to a condition characterized by high levels of energy and mental resilience during the job experience, associated with the intention to make efforts in a voluntary and persistent way, even when faced with adversity. Dedication is characterized by the perception of work as an experience which has meaning, with subsequent feelings of enthusiasm, inspiration, and pride. Lastly, absorption indicates the sensation of feeling completely focused on one's work, which one enjoys and finds it hard to detach from [54].

Recent meta-analyses on the antecedents and consequences of work engagement have made it possible to verify the relationships hypothesized by the job demands–resources theoretical model. In particular, Halbesleben's study [57] has highlighted how job demands are effectively associated in a negative way with work engagement, particularly with regard to work-life conflict, work from home, and workload. Conversely, resources are positively correlated with work engagement, specifically about aspects such as social support, autonomy and control of one's own work actions, feedback and social relations, organizational climate, and individual resources, such as self-efficacy and optimism. Lastly, what was also highlighted is how work engagement has a positive influence on certain outcomes such as work investment, performance, and health, and a negative influence on turnover [57]. Furthermore, as stated by Tims et al. [58], work engagement seems to be strengthened by proactive behaviors such as job crafting, both when it is practiced at an individual level and at a group level. Therefore, some job aspects can activate job-crafting behavior, which has a positive association with motivational processes and work engagement [59,60].

### 2.1.4. Psychological Well-Being

The conceptualizations of psychological well-being have largely been influenced over the years by the two primary concepts of positive functioning. On the one hand, well-being and happiness coincide with a balance between positive and negative affect; on the other hand, which is mostly highlighted by sociologists, satisfaction is the key to well-being. However, current literature views well-being as a multi-dimensional construct, which is closely correlated with mental health. Subsequent studies essentially define the existence of two macro-categories of thought: on the one hand, a perspective that combines well-being with happiness (the hedonic school of thought) and, on the other, an approach that assimilates it with human potential, which can lead to positive functioning in life when fully realized (a eudemonic thought process) [61]. The hedonic perspective defines well-being as an element, which encompasses happiness and experiencing pleasant emotions. Well-being, therefore, becomes a multi-dimensional concept which encompasses perceptions such as assessments of life, in general, in emotional terms, and the presence of positive effects and the absence of negative effects. A fundamental characteristic of the hedonic approach to subjective well-being is the key focus on the dimension linked to the presence of positive emotions and the absence of negative emotions [62]. On the other hand, the eudemonic approach views well-being from a purely psychological perspective, which is mainly self-realization in the sense of full expression of the potential, resources, and individual predispositions to build meanings and reach objectives [61,63]. Viewed in these terms, the connection between the well-being of a single individual and his/her reference context becomes ambiguous: examining well-being is no longer synonymous with pleasure; it is expressed when the ability to follow important objectives and mobilize resources

for the individual and society is achieved. Other dimensions that are strictly connected with well-being are an increase in individual abilities, autonomy, and social competencies and the role taken on by interpersonal relations in the promotion of individuals and the community [64,65]. In this theoretical framework, well-being can be operationally measured using a psychological and social dimension. Psychological well-being indicates a multi-dimensional construct that includes self-acceptance, positive relationships with others, autonomy, a sense of confidence and competence, having a purpose in life, and a desire for personal growth [61,63]. Different studies have made it possible to establish how different variables interact to improve the psychological well-being of workers. More specifically, as highlighted by the job demands–resources model, aspects such as job autonomy [17], social support [6], feedback from managers, proactive skills [66], a sense of self-efficacy [15], and all the elements which define job resources [33] have proven to be strong predictors of well-being. On the contrary, demands (such as an excessive workload [16], difficulty in creating a healthy work–life balance [16], and job insecurity [37] undermine the quality of organizational life.

### 2.1.5. Emotional Exhaustion

Today, managing psychological stress generated by work conditions has become more important. About 25% of Italian workers report a low level of mental well-being, which is strongly indicative of a significant risk of depression [67]. The consequences of stress can manifest in physiological (headaches, insomnia, and fatigue), psychological (anxiety, frustration, and dissatisfaction), and behavioral aspects (absenteeism, presenteeism, and social isolation); however, when the situation becomes chronic, the risk of burnout increases, that is "a condition of emotional exhaustion dictated by the feeling of not having the resources necessary to cope with work demands" [6]. Among the different theories of burnout, Maslach et al. [68] developed a multi-dimensional theory in which three components are hypothesized: emotional exhaustion, disaffection (or cynicism), and reduced personal efficacy. Emotional exhaustion is the main manifestation of burnout, and it refers to the feeling of having used up all the physical–psychological energies required to deal with work demands. Depersonalization refers to the emotional and cognitive detachment that occurs in relationships with users/patients from a dehumanized and cynical perception of them. Lastly, diminished professional realization is the negative assessment of one's work results and the ability to deal with work demands. Over the years, several discussions on the content and validity of this conceptualization have taken place [68–70], especially as burnout syndrome is experienced predominantly in caregiving professions. This is why recent literature considers the dimension of emotional exhaustion as transversal to all professions, not just caregiving ones. Over the years, many studies have investigated the antecedents of burnout, which are, for the most part, the same as those of work stress. In general, it is possible to distinguish between individual and work factors. The personal factors include age (workers under the age of 30 with limited work experience indicate a higher incidence of burnout—[71]; gender (women in general report higher exhaustion levels—[68]); and some personality features, such as personal resources (personal efficacy, optimism, self-esteem, etc.). The work factors that increase stress include elevated work demands, such as emotional work, temporary pressure, workload, conflict, and ambiguity of the role [33,72]; some leadership behaviors, such as lack of respect, excessive demands, absence of feedback, techno-stress [73], and continuous criticism [74]. Some work resources, however, have a protective role. These are autonomy, social support, and recognition [33,72].

### 2.2. Objectives and Hypotheses

By virtue of the above-described theoretical framework, and in line with the classification of the variables under study according to the dimensions of demands and resources within the JD-R model, the hypotheses of the study are as follows:

**H₁**: *Job insecurity (job demand) negatively impacts job crafting, work engagement, and psychological well-being, while it is positively associated with emotional exhaustion.*

**H₂**: *Job crafting positively affects work engagement, psychological well-being, and employability, and negatively affects emotional exhaustion.*

**H₃**: *Age negatively impacts job crafting.*

**H₄**: *The above-mentioned relationships are not significantly different depending on the contract type (temporary/permanent) and type of organization (private/public).*

The hypothesized structural relationships are reported graphically in Figure 1.

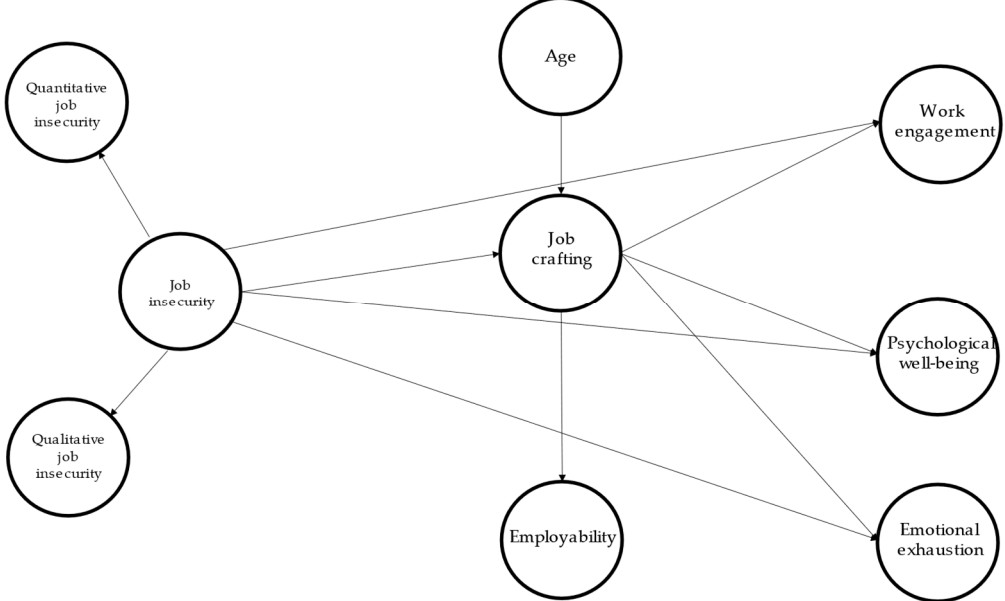

**Figure 1.** The relationships of the structural model between the latent variables hypothesized in the study.

### 3. Materials and Methods

*3.1. Latent Variables Approach: PLS-SEM and CB-SEM*

The statistical procedure used to assess the assumptions of both studies was established for the modeling of latent variables using PLS-SEM [75–80]. The latent variable approach, widely applied in certain fields such as psychology [81,82], aims to measure relationships between latent variables or dimensions that have the characteristic of reflecting constructs that are not immediately observable. Using two separate regression models, it is possible to measure latent factors using manifest indicators (items on a questionnaire, for example) and verify the relationship between these dimensions. The PLS-SEM is a structural equation modeling (SEM) which, unlike the normal parametric approach, is based on the matrices of co-variance (CB-SEM), and its objective is to estimate parameters to maximize the variance of endogenous constructs and relative indicators through relationships between conceptual variables which are eminently non-observable. The non-parametric structural equation models allow the release of some criteria necessary to carry out SEMs, such as multi-variate normality tests, considering single-item latent factors, and including large quantities of data in the study. For this last affirmation, we need to add an integration: while one of the most frequent motives adopted using the PLS-SEM is the scarce sample dimension, different studies allowed us to establish that the estimated bias of the parameters does not differ significantly in terms of an increase in the sample size [79,83]. The biggest difference between PLS-SEM and CB-SEM exists in common factor models with datasets above 10,000 observations, as highlighted by [84] and proposed in Table 1. However, the data simulation carried out at 100, 250, 500, 1000, and 10,000 observations allowed us to affirm that the distortion of the estimates of the parameters is only 0.02

between PLS and SEM. This is significantly different when we want to perform analyses using composite model designs. The difference in the estimate of the parameters between the two techniques in this case increased up to the values of approximately 0.72.

**Table 1.** Difference in the estimate of parameters in terms of using common factor models (to the left) and composite models (to the right).

| Design Factor | | | Mean Absolute Error (MAE) | | Design Factor | | | Mean Absolute Error (MAE) | |
|---|---|---|---|---|---|---|---|---|---|
| Observation | Group | Loadings | PLS | CBSEM | Observation | Indicators | Weights | PLS | CBSEM |
| 100 | 2 indicators | Mixed | 0.11 | 0.13 | 100 | 2 | Equal | 0.07 | 0.84 |
| | 4 indicators | | 0.09 | 0.10 | | 4 | | 0.07 | 0.74 |
| | 6 indicators | | 0.08 | 0.08 | | 6 | | 0.07 | 0.63 |
| | 8 indicators | | 0.08 | 0.08 | | 8 | | 0.07 | 0.52 |
| | Loadings: 0.5 | Equal | 0.13 | 0.16 | | 2 | Unequal | 0.07 | 0.61 |
| | Loadings: 0.7 | | 0.09 | 0.10 | | 4 | | 0.08 | 0.92 |
| | Loadings: 0.9 | | 0.06 | 0.07 | | 6 | | 0.07 | 0.54 |
| | Loadings: 0.5/0.9 | Unequal | 0.08 | 0.08 | | 8 | | 0.07 | 0.48 |
| 250 | 2 indicators | Mixed | 0.10 | 0.09 | 250 | 2 | Equal | 0.05 | 0.81 |
| | 4 indicators | | 0.08 | 0.06 | | 4 | | 0.04 | 0.81 |
| | 6 indicators | | 0.06 | 0.05 | | 6 | | 0.05 | 0.73 |
| | 8 indicators | | 0.06 | 0.05 | | 8 | | 0.04 | 0.57 |
| | Loadings: 0.5 | Equal | 0.12 | 0.10 | | 2 | Unequal | 0.05 | 0.54 |
| | Loadings: 0.7 | | 0.07 | 0.06 | | 4 | | 0.05 | 0.82 |
| | Loadings: 0.9 | | 0.04 | 0.04 | | 6 | | 0.05 | 0.58 |
| | Loadings: 0.5/0.9 | Unequal | 0.06 | 0.05 | | 8 | | 0.05 | 0.59 |
| 500 | 2 indicators | Mixed | 0.10 | 0.06 | 500 | 2 | Equal | 0.03 | 0.90 |
| | 4 indicators | | 0.07 | 0.04 | | 4 | | 0.03 | 0.81 |
| | 6 indicators | | 0.06 | 0.04 | | 6 | | 0.03 | 0.82 |
| | 8 indicators | | 0.05 | 0.04 | | 8 | | 0.03 | 0.75 |
| | Loadings: 0.5 | Equal | 0.12 | 0.07 | | 2 | Unequal | 0.03 | 0.55 |
| | Loadings: 0.7 | | 0.07 | 0.04 | | 4 | | 0.03 | 0.62 |
| | Loadings: 0.9 | | 0.03 | 0.03 | | 6 | | 0.03 | 0.60 |
| | Loadings: 0.5/0.9 | Unequal | 0.05 | 0.03 | | 8 | | 0.02 | 0.56 |
| 1000 | 2 indicators | Mixed | 0.09 | 0.04 | 1000 | 2 | Equal | 0.02 | 0.98 |
| | 4 indicators | | 0.06 | 0.03 | | 4 | | 0.02 | 0.80 |
| | 6 indicators | | 0.05 | 0.03 | | 6 | | 0.02 | 0.95 |
| | 8 indicators | | 0.05 | 0.03 | | 8 | | 0.03 | 0.88 |
| | Loadings: 0.5 | Equal | 0.12 | 0.05 | | 2 | Unequal | 0.02 | 0.55 |
| | Loadings: 0.7 | | 0.06 | 0.03 | | 4 | | 0.02 | 0.65 |
| | Loadings: 0.9 | | 0.03 | 0.02 | | 6 | | 0.02 | 0.65 |
| | Loadings: 0.5/0.9 | Unequal | 0.05 | 0.02 | | 8 | | 0.01 | 0.65 |
| 10,000 | 2 indicators | Mixed | 0.09 | 0.01 | 10,000 | 2 | Equal | 0.01 | 1.34 |
| | 4 indicators | | 0.06 | 0.01 | | 4 | | 0.01 | 0.74 |
| | 6 indicators | | 0.04 | 0.01 | | 6 | | 0.01 | 1.72 |
| | 8 indicators | | 0.04 | 0.01 | | 8 | | 0.01 | 1.22 |
| | Loadings: 0.5 | Equal | 0.11 | 0.01 | | 2 | Unequal | 0.01 | 0.66 |
| | Loadings: 0.7 | | 0.06 | 0.01 | | 4 | | 0.01 | 0.51 |
| | Loadings: 0.9 | | 0.02 | 0.01 | | 6 | | 0.01 | 1.02 |
| | Loadings: 0.5/0.9 | Unequal | 0.04 | 0.01 | | 8 | | 0.01 | 0.58 |
| Total | | | 0.07 | 0.05 | Total | | | 0.04 | 0.76 |

Therefore, in general, in PLS-SEM, the latent constructs hypothesized are composite dimensions—not common factors—which can be measured by indicators manifested using the reflexive method (variations in the latent variables cause direct variations in the indicators assigned) and formative method (fluctuations in one or more manifested indicators cause fluctuations in the latent variable [85]). The use of PLS-SEM in academic research has highlighted an exponential increase in the last ten years, even though studies using this technique have mostly focused on the financial and managerial contexts of the workforce for companies. The presuppositions from which the exploratory analysis is derived, which can be carried out with or without PLS-SEMs, can be experimented even in a psychological sphere, where very often latent constructs are hypothesized but are not immediately observable, and the samples are rarely probabilistic. These are the reasons why this research was conducted using this technique.

### 3.2. Participants

The survey was conducted on 641 working professionals and was balanced in terms of gender. A total of 52.7% were women (338); 47.3% were men (303). From an educational perspective, most of the sample claimed to have medium-high titles; specifically, 53% had a high school diploma (340) and 17.6% had a four-year degree or a third-year degree. In terms of marital status, there was a balance between people who were married/co-habitating (49.6%, 318 individuals) and single (44%, 286 individuals). In terms of their professional circumstances, 59.1% of all people interviewed had a permanent contract (379); 25.3% had a temporary contract (162); and 15.6% were freelancers. In total, 52.6% of the workers performed their jobs in a private organization (337) and 47.4% in a public organization (304); 54.1% (347) did not have children and 45.9% did (294). The average age of the participants was 38.6 years; the age range was 16 to 70 years. Table 2 reports the principal descriptive characteristics of the sample.

**Table 2.** Principal descriptive characteristics of the sample.

| Sample: 641 Workers | Mean Age = 38.6 |
|---|---|
| Gender | 52.7% Women |
| | 47.3% Men |
| Educational level | 53% High school diploma |
| | 17.6% Four-year or third-year degree |
| Marital status | 49.6% Married/Co-habitating |
| | 44% Single |
| Professional characteristics | 59.1% Permanent contract |
| | 15.6% Freelancer |
| | 52.6% Private organization |
| | 47.4% Public organization |
| Children | 54.1% With children |
| | 45.9% No children |

### 3.3. Data Collection and Data Analysis

The questionnaire was filled out from November 2018 to March 2019 via a link received online. The convenience sampling method was adopted. Research participants, who were assured that they could leave the study at any point, were provided with all the key information pertaining to the research; all procedures complied with regulations about privacy.

The data were collected using a quantitative questionnaire consisting of a series of variables measuring constructs, identified after a thorough literature search to confirm

their psychometric and epistemological validity. Therefore, the identified questionnaires from which the items were extracted met the appropriate criteria of methodological and epistemological consistency. Participants were recruited using non-probabilistic snowball sampling, which involved requesting the questionnaire to be sent to those who completed it in a chain process. Prior to answering the questionnaire, within which appropriate guarantees of anonymity and data aggregation were ensured, the participants provided informed consent. Respondents were given the opportunity to withdraw from completing the questionnaire at any time.

The data were analyzed anonymously, with no possibility of tracing them back to the individual participant under any circumstances. After appropriate descriptive, correlation, reliability ($\alpha$, $\rho$, composite reliance, average variance extracted), and discriminant validity analyses (HTMT matrix), a general PLS-SEM model was performed. Subsequently, a multi-group comparison was performed to understand possible differences regarding organizational sector and contract type.

*3.4. Measures*

The hypotheses under investigation were verified by using scales validated in the literature. Moreover, each variable was epistemologically considered, according to the literature, within the category proposed by the job demands–resources model. The job demands–job resources model, as mentioned above, is an extremely flexible but specific conceptual paradigm that can be adapted to different application contexts. In this regard, a variety of studies spread its empirical concretization to multiple organizational contexts, such as schools [86], disability [87], universities [88,89], and work environments in general [90]. For these reasons, these sources agree in defining the JD-R model as epistemologically appropriate for the purposes of this study are as follows:

- Job insecurity: This is considered a second-level latent factor based on the quantitative and qualitative insecurity scales, measured using the repeated indicators approach. The quantitative scale of insecurity at work is made up of five items [91], with Cronbach's $\alpha$ = 0.83 and McDonald's $\omega$ = 0.85. An example of an item is "*I'm scared of getting fired*". According to [92], job insecurity is deemed to be a demand.

    The qualitative scale of insecurity at work [38], made up of four items, reveals reliance indexes equal to Cronbach's $\alpha$ = 0.78 and McDonald's $\omega$ equal to 0.79. An example of an item is "*I'm worried about my career progression in my organization*". The response scale ranged from 1 = Completely disagree to 5 = Completely agree. The total reliance of the factor was $\alpha$ = 0.82 and $\omega$ = 0.82.

- Job crafting: Only the positive dimensions of these constructs were considered [93]; more specifically, an increase in structural resources, an increase in social resources, and an increase in challenging demands. The scale, made up of nine items [94], indicates appropriate reliance, with $\alpha$ = 0.91 and $\omega$ = 0.92. According to [9,16], job crafting is included as a mediator in the model.

    The examples of items for each sub-section are "When an interesting project is put forward, I actively put myself forward to participate", "I ask colleagues of the groups I'm involved in to give me instructions and suggestions to improve my work", and "I try to develop professionally". The response scale ranged from 1 = Never to 6 = Always.

- Employability: This refers to six items from Berntson and Marklund [45], in which Cronbach's $\alpha$ and McDonald's $\omega$ are identical at 0.84. An example of an item is "*I'm actively committed to increasing my "spendability" on the work market*", with a response scale ranging from 1 = Completely disagree to 5 = Completely agree. Employability, as a proactive behavior, is considered positively associated with job crafting, as suggested by [95].

- Work engagement: Three items [54] were considered, with one item for each of the three different parameters, which are dedication, absorption, and vigor. Reliance was indicated with Cronbach's $\alpha$ and McDonald's $\omega$ equal to 0.86. An example of

an item is "*I'm happy when I'm busy at work*", with a response scale from 1 = Never to 6 = Always. Work engagement is regarded as a positive outcome linked to the motivational process, as assumed by [57,96].

- Psychological well-being: This parameter was extracted from the questionnaire by Petrillo et al. [61] with six items in total. The internal coherence of the scale appears to be respected as both Cronbach's α and McDonald's ω were equal to 0.90. Examples of items are "*Do you feel you have had experiences that have helped you to grow and become a better person?*" and "*Do you feel able to think or express your ideas and opinions?*", with a response scale ranging from 0 = Never and 5 = Every day. The constructs, such as work engagement, were incorporated into the positive outcomes associated with the motivational process, as various studies showed [97,98].
- Emotional exhaustion: Four items from Kristensen et al. [99] were considered. An example of an item is "*I feel exhausted at the end of a workday*". Cronbach's α and McDonald's ω were equal to 0.90, with a response scale ranging from 1 = Never to 6 = Always. Emotional exhaustion, as a dimension of burnout, is an important part of the processes associated with health impairment according to the JD-R model, as shown in several previous studies [96].

## 4. Results

The correlations indicate good correspondence between the relationships theorized using the JD-R Model and the retrieved outputs (see Table 3). For example, the relationship between both quantitative and qualitative insecurity, which is considered a demand according to the conceptual model, and psychological well-being, which is a process connected to motivation, shows a negative and significant correlation ($r_1 = -0.222$, <0.001 e $r_2 = -0.164$, <0.001). A general analysis shows that psychological well-being is positively and significantly correlated with employability ($r_3 = 0.328$, <0.001), work engagement ($r_4 = 0.493$, <0.001), and job crafting ($r_5 = 0.388$, <0.001), whereas job insecurity strongly is correlated with emotional exhaustion ($r_6 = 0.377$, <0.001). Age does not present biunivocal relations worth mentioning in relation to psychological well-being, while its increase corresponds to a decrease in both emotional exhaustion and job crafting. Lastly, there is a strongly positive and significant association between job crafting and employability ($r_7 = 0.536$, <0.001).

**Table 3.** Correlation matrix between the researched variables.

| | 1 | 2 | 3 | 4 | 5 | 6 | 7 | 8 |
|---|---|---|---|---|---|---|---|---|
| 1. Psychological well-being | | | | | | | | |
| 2. Emotional Exhaustion | −0.325 *** | | | | | | | |
| 3. Employability | 0.328 *** | −0.088 * | | | | | | |
| 4. Age | 0.059 | −0.116 ** | −0.110 ** | | | | | |
| 5. Job Crafting | 0.388 *** | −0.132 ** | 0.536 *** | −0.073 | | | | |
| 6. Job insecurity (Qualitative) | −0.222 *** | 0.373 *** | −0.009 | −0.030 | −0.084 * | | | |
| 7. Job insecurity (Quantitative) | −0.164 *** | 0.259 *** | 0.068 | −0.180 *** | −0.056 | 0.368 *** | | |
| 8. Job insecurity | −0.229 *** | 0.377 *** | 0.034 | −0.121 ** | −0.089 * | 0.810 *** | 0.843 *** | |
| 9. Work engagement | 0.493 *** | −0.411 *** | 0.381 *** | 0.041 | 0.412 *** | −0.266 *** | −0.069 | −0.202 *** |

Note: *** < 0.001, ** < 0.01, * < 0.05.

The analyses were carried out using SMARTPLS software [100]. The model hypothesized was validated using the bootstrap methodology with 5000 re-samples, with relative analyses of the confidence intervals. The missing data were estimated by substitution using the average. From the perspective of measuring models, all constructs highlight excellent and significant saturations, particularly in the range between 0.78 and 0.85 for the latent psychological well-being dimension; between 0.84 and 0.91 for emotional exhaustion; 0.67 and 0.86 for job crafting; 0.68 and 0.78 for employability; 0.62 and 0.84 for qualitative

professional insecurity; 0.54 and 0.90 for quantitative work insecurity; and 0.88 and 0.88 for professional work.

Even the convergent and discriminating validity is confirmed because, for each construct, the average variance extracted (AVE) exceeds 50% (Table 4) and the monotract–heterotract matrix (Table 5 highlights minor values of 0.90 [77,80,101]).

**Table 4.** Reliability analyses of hypothesized latent constructs.

| | $\alpha$ | $\rho$ | Composite Reliance | Average Variance Extracted (AVE) |
|---|---|---|---|---|
| Psychological well-being | 0.90 | 0.91 | 0.93 | 0.67 |
| Emotional exhaustion | 0.90 | 0.90 | 0.93 | 0.77 |
| Employability | 0.84 | 0.86 | 0.88 | 0.55 |
| Job Crafting | 0.92 | 0.93 | 0.93 | 0.60 |
| Job insecurity (qualitative) | 0.78 | 0.79 | 0.86 | 0.60 |
| Job insecurity (quantitative) | 0.83 | 0.87 | 0.88 | 0.61 |
| Work engagement | 0.86 | 0.87 | 0.91 | 0.78 |

**Table 5.** Heterotract–Monotract Matrix (HTMT) to verify the discriminating validity of the constructs.

| | 1 | 2 | 3 | 4 | 5 | 6 | 7 |
|---|---|---|---|---|---|---|---|
| 1. Psychological well-being | | | | | | | |
| 2. Emotional Exhaustion | 0.34 | | | | | | |
| 3. Employability | 0.39 | 0.15 | | | | | |
| 4. Age | 0.06 | 0.11 | 0.12 | | | | |
| 5. Job Crafting | 0.43 | 0.14 | 0.63 | 0.07 | | | |
| 6. Job insecurity (Qualitative) | 0.18 | 0.42 | 0.13 | 0.12 | 0.12 | | |
| 7. Job insecurity (Quantitative) | 0.19 | 0.30 | 0.16 | 0.19 | 0.13 | 0.44 | |
| 8. Work engagement | 0.56 | 0.46 | 0.48 | 0.04 | 0.48 | 0.23 | 0.14 |

Figure 2 highlights the model with overall sample, its coefficients and significance. Regarding hypothesis $H_1$, the overall model shows how professional insecurity negatively impacts work engagement ($\beta_1 = -0.13$ [$-0.20$; $-0.06$], $p < 0.000$) and psychological well-being ($\beta_2 = -0.17$ [$-0.24$; $-0.09$], $p < 0.000$), while emotional exhaustion goes up ($\beta_3 = 0.36$ [0.29; 0.44], $p < 0.000$). At the same time, however, job insecurity does not seem to impact the propensity (or lack thereof) to display proactive job-crafting behaviors ($\beta_4 = -0.07$ [$-0.16$; 0.01], $p = 0.105$, ns). The $H_2$ hypothesis is confirmed as job crafting in its mediator role, positively influencing both work engagement ($\beta_5 = 0.42$ [0.34; 0.51], $p < 0.000$) and psychological well-being ($\beta_6 = 0.39$ [0.31; 0.47], $p < 0.000$), while it has a protective role in emotional exhaustion ($-0.11$ [$-0.18$; $-0.03$], $p < 0.05$). Furthermore, job crafting seems to trigger certain employability behaviors, as the two constructs are positively and significantly associated ($\beta_7 = 0.59$ [0.53; 0.65], $p < 0.000$). Lastly, hypothesis $H_3$ appears to be validated (notwithstanding the fact that the confidence level is very close to 0) because age influences job crafting; more specifically, the older the workers, the less likely they are to develop proactive behavior to adapt their professional tasks to personal needs ($\beta_8 = -0.10$ [$-0.16$; 0.00], $p < 0.05$).

To investigate hypothesis $H_4$, the models were also calculated, categorizing the subjects by contract type (permanent and temporary, see Figure 3) and organization (public and private, see Figure 4); then, a multi-group analysis was carried out to identify significant differences between the coefficients (see Table 6). The results revealed a substantial overlap of the coefficient models between temporary and permanent workers.

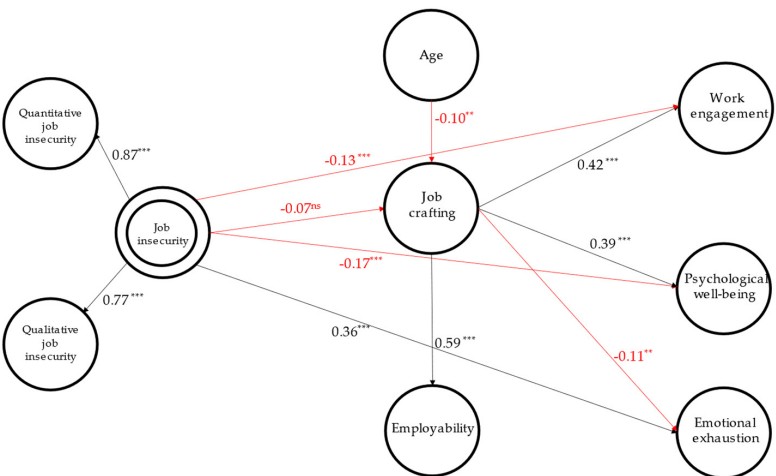

**Figure 2.** The final structural model with the complete sample. Note: *** < 0.001, ** < 0.01. Red lines means negative statistical relationships.

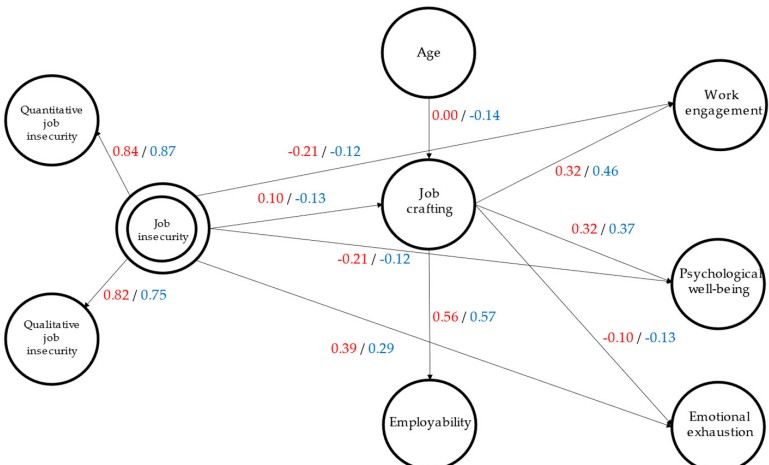

**Figure 3.** Multi-group analysis by contract type (temporary/permanent) carried out using the permutation algorithm. In red are the coefficients for temporary workers and in blue are the coefficients for permanent workers.

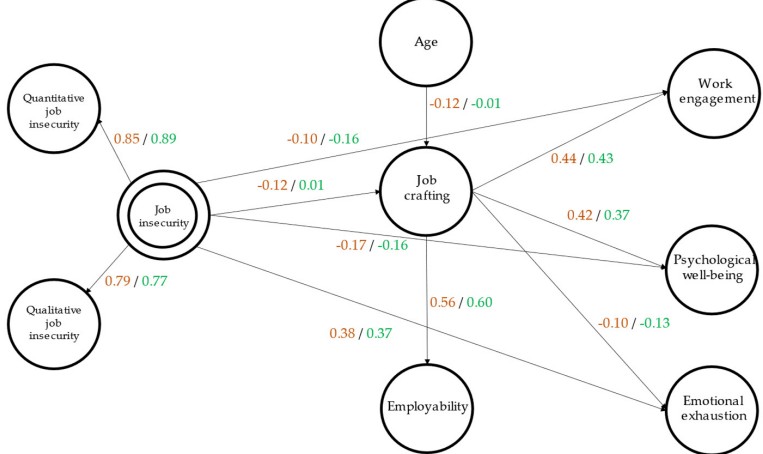

**Figure 4.** Multi-group analyses by organization type (private/public) carried out using the permutation algorithm. In orange are the coefficients for workers in private organizations; in green are the coefficients for public organizations.

**Table 6.** Multi-group analyses using permutations of the coefficients of the models of permanent and temporary workers.

|  | TEMPORARY | PERMANENT | TEMP—PERM | $p$ |
|---|---|---|---|---|
| Age → Job Crafting | 0.00 | −0.14 | 0.14 | 0.123 |
| Job Insecurity → Psychological Well-Being | −0.21 | −0.12 | −0.09 | 0.368 |
| Job Insecurity → Emotional Exhaustion | 0.39 | 0.29 | 0.10 | 0.290 |
| Job Insecurity → Job Crafting | 0.10 | −0.14 | −0.24 | 0.050 |
| Job Insecurity → Qualitative Job Insecurity | 0.82 | 0.75 | 0.07 | 0.188 |
| Job Insecurity → Quantitative Job Insecurity | 0.84 | 0.87 | −0.03 | 0.218 |
| Job Insecurity → Work Engagement | −0.21 | −0.12 | −0.09 | 0.348 |
| Job Crafting → Psychological Well-Being | 0.32 | 0.37 | −0.05 | 0.667 |
| Job Crafting → Emotional Exhaustion | −0.10 | −0.13 | 0.03 | 0.701 |
| Job Crafting → Employability | 0.56 | 0.57 | −0.01 | 0.911 |
| Job Crafting → Work Engagement | 0.32 | 0.46 | −0.14 | 0.199 |

More specifically, regarding the distinction between temporary and permanent contracts, as theorized by the JD-R model [13,35], job crafting has a protective effect on emotional exhaustion as it is negatively associated with ($\beta_{TEMP} = -0.10/\beta_{PERM} = -0.13$). Conversely, it leads to better work engagement ($\beta_{TEMP} = 0.32/\beta_{PERM} = 0.46$) and psychological well-being ($\beta_{TEMP} = 0.32/\beta_{PERM} = 0.37$). In addition, job crafting has a positive effect on the employability of workers irrespective of the type of contract ($\beta_{TEMP} = 0.56/\beta_{PERM} = 0.57$). Lastly, age seems to impact the frequency of job crafting behaviors in a negative way only for temporary workers. The impact of age on job crafting is non-existent among temporary workers ($\beta_{TEMP} = 0.00/\beta_{PERM} = -0.14$). Therefore, the multi-group analysis carried out using the algorithm of [102] does not reveal any significant differences between the coefficients, except for in the relationship between work insecurity and job crafting; for temporary jobs, insecurity seems to have a positive impact on pro-active behaviors by increasing them, while the opposite is true for permanent workers ($\beta_{TEMP} = 0.10/\beta_{PERM} = -0.14$). Lastly, job insecurity seems to worsen psychological well-being ($\beta_{TEMP} = -0.21/\beta_{PERM} = 0.12$) and work engagement ($\beta_{TEMP} = -0.21/\beta_{PERM} = -0.12$), irrespective of the insecurity of the contract, even though it is more noticeable among temporary workers.

Regarding differentiation based on the type of organization (private or public, see Figure 4), the multi-group analysis carried out using the permutation algorithm does not highlight significant differences between the coefficients (see Table 7). In descriptive terms, the most significant discrepancies can be identified regarding the impact of age on job crafting behaviors, which is negatively higher in private organizations and essentially irrelevant in public organizations ($\beta_{PRIV} = -0.12/\beta_{PUB} = -0.01$), and in terms of the impact of job insecurity on job crafting ($\beta_{PRIV} = -0.12/\beta_{PUB} = 0.01$), it is negatively higher in private organizations, whereas in public organizations, its impact is practically non-existent. Therefore, job insecurity increases the risk of emotional exhaustion irrespective of the organization ($\beta_{PRIV} = 0.38/\beta_{PUB} = 0.37$), while it reduces both psychological well-being ($\beta_{PRIV} = -0.17/\beta_{PUB} = -0.16$) and work engagement ($\beta_{PRIV} = -0.10/\beta_{PUB} = -0.16$). Job crafting, on the other hand, irrespective of the organization, improves motivational processes, such as psychological well-being ($\beta_{PRIV} = 0.42/\beta_{PUB} = 0.37$), work engagement ($\beta_{PRIV} = 0.44/\beta_{PUB} = 0.43$), and active behavior to improve professional employability ($\beta_{PRIV} = 0.58/\beta_{PUB} = 0.60$).

Table 8 shows the effect size measurements of the hypothesized relationships between the latent variables, delving into the effect of exogenous on endogenous variables to determine changes in the $R^2$ value when specific exogenous determinants are excluded from the model [103]. Accordingly, if $f^2$ is less than 0.15, the effect is small, if it is between 0.15 and 0.35, it is medium, and if it exceeds 0.35, it is large [104]. Based on the obtained values, most of the relationships show a small effect size (50.9%), followed by 27.3% large effects and 21.8% medium effects.

**Table 7.** Multi-group analyses using permutations of the coefficients of the models in private and public organizations.

|  | PRIVATE | PUBLIC | PRIVATE—PUBLIC | *p* |
|---|---|---|---|---|
| Age → Job Crafting | −0.12 | −0.01 | −0.11 | 0.169 |
| Job Insecurity → Psychological Well-Being | −0.17 | −0.16 | −0.00 | 0.961 |
| Job Insecurity → Emotional Exhaustion | 0.38 | 0.37 | 0.01 | 0.898 |
| Job Insecurity → Job Crafting | −0.12 | 0.01 | −0.12 | 0.145 |
| Job Insecurity → Qualitative Job Insecurity | 0.79 | 0.77 | 0.02 | 0.486 |
| Job Insecurity → Quantitative Job Insecurity | 0.85 | 0.89 | −0.04 | 0.066 |
| Job Insecurity → Work Engagement | −0.10 | −0.16 | 0.06 | 0.461 |
| Job Crafting → Psychological Well-Being | 0.42 | 0.37 | 0.05 | 0.563 |
| Job Crafting → Emotional Exhaustion | −0.10 | −0.13 | 0.03 | 0.501 |
| Job Crafting → Employability | 0.58 | 0.60 | −0.02 | 0.851 |
| Job Crafting → Work Engagement | 0.44 | 0.43 | 0.01 | 0.91 |

**Table 8.** Effect size of structural coefficients between latent variables.

|  | | | MODELS | | |
|---|---|---|---|---|---|
|  | OVERALL | PRIVATE | PUBLIC | TEMPORARY | PERMANENT |
| Age → Job Crafting | small | small | small | small | small |
| Job Insecurity → Psychological Well-Being | small | small | small | small | small |
| Job Insecurity → Emotional Exhaustion | medium | medium | medium | medium | small |
| Job Insecurity → Job Crafting | small | small | small | small | small |
| Job Insecurity → Qualitative Job Insecurity | large | large | large | large | large |
| Job Insecurity → Quantitative Job Insecurity | large | large | large | large | large |
| Job Insecurity → Work Engagement | small | small | small | small | small |
| Job Crafting → Psychological Well-Being | medium | medium | medium | small | medium |
| Job Crafting → Emotional Exhaustion | small | small | small | small | small |
| Job Crafting → Employability | large | large | large | large | large |
| Job Crafting → Work Engagement | medium | medium | medium | small | medium |

## 5. Discussion

The research carried out by this study highlights the role of job crafting as a promoter of well-being and change in companies. The study aimed to verify, within the framework of the job demands–resources model, the mediator effect of proactive behaviors to modify work in the relationship between a professional demand, i.e., professional insecurity, and outcomes connected to motivational processes, such as work engagement and psychological well-being, and mechanisms connected to compromising health and energy, such as emotional exhaustion. Furthermore, the objective of the research was to focus on analyzing the effect of age on job crafting and the latter on the active willingness of workers to improve their employability. The relationships born from this model confirm what was hypothesized by the job demands–job resources, highlighting a positive association between job insecurity and emotional exhaustion [42], and a negative relationship between job insecurity and work engagement and psychological well-being [37].

Therefore, considering the entire sample, insecurity represents job demand. Furthermore, in this sense, it is interesting to note that job insecurity, considering the subjects aggregated by organization and contract type, does not have a significant effect on job crafting; it does not modify workers' propensity to manage their roles and tasks in line with their personal outcomes [32]. On the contrary, job crafting, as widely investigated in the literature, proves to be an important strategy to improve work engagement and psychological well-being and, at the same time, to limit processes such as emotional exhaustion [15–17]. In addition, as highlighted in studies on diversity management in organizations, age effectively negatively impacts job crafting [25]; younger employees tend to be more likely to implement proactive management behavior in managing their work.

Lastly, job crafting seems to have influenced employability, thus representing a powerful means to be used by organizations aimed at improving well-being and life quality [8].

The same dynamic and polarity of the coefficients are identifiable even by executing models individually for the type of professional contract (temporary and permanent) and organizational type (public and private), except for the relationship between job insecurity and job crafting amongst workers with a temporary contract. In this case, job insecurity positively activates job crafting behavior, thus representing a sort of activating process and "challenging demand" [105]. At the same time, when it comes to temporary workers, age does not seem to influence job-crafting behavior; whereas, for people with professional stability, the tendency follows the one hypothesized in the literature.

Thus, this research poses some critical issues that need to be discussed. Job insecurity, especially during the last pandemic period, has increased disproportionately (see, as an example, the definition of technostress, which has at its core dimensions such as techno-complexity and techno-uncertainty, factors associated with job insecurity [73]) and is an extremely important risk factor in determining negative outcomes such as stress. The perception of job insecurity, therefore, plays a key role in fostering healthy organizational contexts [23,24]. In turn, however, this factor does not affect the implementation of proactive job crafting behaviors, which is why it is plausible to think that the latter, which has positive effects on employability, psychological well-being, and work engagement and decreases emotional exhaustion, can be counted among the sustainable competencies to be developed at an individual level to have an impact on the quality of working life. Job crafting, therefore, could reveal itself as a personal tool that enhances sustainability with a focus on a human dimension that, as the authors of [106] argue, is less debated than the environmental, economic, and technological dimensions, but it strongly accounts for successful innovation [28]. Indeed, in a period in which the need for organizations to equip themselves with sustainable equipment is highly argued, the aim is to achieve a triple bottom line of economic, environmental, and human performance (through human sustainability).

Human sustainability addresses the core aim of improving society's well-being through concrete practices that enhance the quality of human life [107]. The means by which this objective is to be achieved is through the implementation of specific skills, methods, and strategies used to improve human life and collective well-being. From a work perspective, human sustainability refers to the development of skills and knowledge to improve the quality of human life. Accordingly, this study intends to provide indications on how acting on an employee's proactive ability to mold the activities in line with needs can indirectly benefit sustainability, with an impact on the quality of professional and, consequently, personal life.

The discussion, however, cannot be free from referring to the role of the organizational context in the quality of professional life. Although the research predominantly focuses on the sustainable role of job crafting, it is worth considering how certain HR characteristics may influence the development of these behaviors [108]. Numerous studies [109,110] have pointed out that functional HR practices can ensure a better psychological climate and thus influence the creation of healthy organizations [24,111]. Therefore, it is also necessary to include this dimension of influence.

## 6. Conclusions

This study provides reflections that can be useful in planning organizational interventions with the goal of improving well-being and quality of life, but, at the same time, this study presents limitations that are important to take into consideration. First, the sample was carried out in a non-probabilistic way. More specifically, a convenience procedure that forces people to be careful when it comes to generalizing results was adopted. Second, the assessments and measurements of the constructs are self-reports by nature; therefore, they are subject to oscillations and do not have objective characteristics. Third, the statistical methodology adopted, despite having advantages in that it adopts some typical assumptions of the parametric methods (quantity of the samples, normal distribution, and latent variables with single items [75,78,80] present causal connections readable in an exploratory form); therefore, further research which may confirm (or disprove) the results is required.

With these affirmations in mind, this study allows for application scenarios and reflections of great importance in terms of health organizations and positive psychology [112]. As a matter of fact, from Seligman's prospective organization, in which great importance is given to the worker's 'positive' factors which can promote better management strategies for the circumstances of one's profession [112], what emerges is how job crafting effectively has a protective role towards negative outcomes, such as emotional exhaustion, and an activating role with advantageous consequences, such as work engagement and psychological well-being [15,17,93]. Furthermore, considering the entire sample, it was possible to highlight how job insecurity, which is included among aspects of insecurity that negatively impact motivational processes, does not significantly impact the ability to execute proactive behavior [37]. Therefore, job crafting seems to be a useful tool even in situations where workers are unsure of whether they can maintain their workplace. Job crafting also seems to be a powerful action tool to address this insecurity, especially among temporary workers, as the relationship that binds these two constructs is positive. To sum up, for people with temporary contracts, insecurity is not an obstacle to the execution of the bottom-up strategy; on the contrary, it is an incentive to further modify the intangible limits of their job, unlike people who have a permanent contract [38]. In this sense, insecurity seems to be a challenging demand, which is a trigger that can increase behavior connected to personal and professional development [32]. Furthermore, analyses on individual models do not highlight significant differences stemming from the organizational type; therefore, these relations, which follow the hypotheses developed by the JD-R model, are robust irrespective of where they are applied [113]. Lastly, the positive role of job crafting materializes even when, as indicated in the data collected from the studies, it helps to promote active behaviors on behalf of the workers to improve their employability [52], an essential element that can allow the creation of sustainable organizations and competencies [23,24].

## 7. Limitations

The study has limitations that need to be considered. Within the study sample, there are socio-demographic characteristics that were not controlled for in the research design; hence, the following limitations must be considered. First, the socio-cultural context of the study needs to be reflected upon to generalize the results. The sample is slightly skewed in favor of female workers, which is why it is plausible to consider the existence of influencing processes that were not considered in this study, such as work-life conflict or the presence of gender stereotypes. Moreover, the sample is predominantly characterized by people who are involved in a relationship; the presence of a partner could, therefore, be a protective factor against outcomes such as emotional exhaustion. The prevalence of educational qualifications belongs to workers with a diploma, so it is plausible to consider that extremely high qualifications with consequent cognitive resources or experience were not considered. The same applies to age because the average of the sample is relatively low (38.6), so it is probably not a question of workers who are either new or newly recruited or about to end their work careers. The type of organization they came from, and the type of contract may also have influenced the results, as public companies are generally recognized as having greater job security than private companies. Finally, although the study was carried out using snowball sampling, it is plausible to assume that most of the workers who participated were from Italy. It is well known that work culture, values, and mission differ from country to country, so caution should be exercised in generalizing the results.

From a methodological point of view, the present work includes other limitations that need to be discussed. First, the cross-sectional research design does not allow for the generalization of the results or the substantiation of further causal relationships. Second, the research measures are predominantly self-reported, with associated problems such as a lack of objectivity or phenomena such as social desirability. Finally, the sample, although large, was identified through a non-probabilistic procedure, so this imposes caution in extending the findings.

## 8. Recommendation

By virtue of what is discussed in the conclusions and limitations, the present work calls for a reflection on possible future scenarios and suggestions for confirming the identified relationships. Further studies wishing to explore the themes of this research could use control variables such as gender, age, or professional role to analyze whether the relationships between the variables change as a function of these characteristics. Moreover, possible future declinations could envisage more structured research designs, considering multiple surveys at different times and of a longitudinal nature, to establish causality in the relationships. Furthermore, the use of more objective measures to measure the constructs under consideration could make this study more structured and objective. Finally, increasing the sample and stratifying it according to a probabilistic perspective could allow the results to be generalized in a more consistent and methodologically correct manner.

## 9. Implications

The above-mentioned preliminary remarks lead to a constructive discussion on how to activate job-crafting behaviors among workers, irrespective of the type of contract or their specific organization of origin. These behaviors—as highlighted in the studies contained in this research—can help professionals and companies create quality environments and foster well-being. Different studies have already highlighted the effects [114] of job crafting interventions even at a longitudinal level, including work performance, work engagement, and balancing demands and resources [114]. These interventions consist of creating workgroups in which, through exercises, the subjects can learn to create empirical strategies to increase structural and social resources, and job demands and, contextually, reduce disruptive demands. Specifically, the program—which lasts six months—foresees consecutive steps: at the beginning, a job analysis is performed to summarize the tasks of each job and it includes how long to dedicate to it, followed by an individual analysis, aimed at identifying subjective strengths, criticalities, and factors which create an obstacle to the correct execution of one's work. Subsequently, there is a comparative phase between the job and the individual analysis, in which workers compare the problematic components of their profession with individual features to detect the features that require strengthening and those that are already suitable. In the fourth and fifth phases, workers are encouraged to reflect on the significant personal changes in their work situations to create structural or social resources, create challenging demands, and reduce disruptive demands, and they are asked to execute such actions for four consecutive weeks, assessing whether these self-produced changes were successful. Lastly, a reflection is made to understand the benefits and criticalities of these job-crafting actions.

The strategic aims of these interventions at an individual level and, consequently, at an organizational level, are clear from the advantages that they generate. Therefore, a reflection on how an action is taken to promote the personal, professional, and cultural growth of workers, focusing on the subjective balance between demands and resources would be ideal [17,114,115]. In conclusion, this study offers several practical implications that both organizations and individual workers should consider. Job crafting, considering the results highlighted both in the literature [116] and in this research, can be considered an important resource for promoting positive outcomes at work and reducing negative outcomes. A rather recent issue related to human resource management precisely concerns the development of employees' potential and their ability to leverage competitive advantage [117].

This theme cuts across various topics, including the competitive advantage of companies, greater worker productivity and commitment, personal and organizational well-being, professional success and career development, and sustainability. Organizations that encourage the development of their employees' resources through intervention paths, including job crafting, work on multiple levels of sustainability. In fact, they not only facilitate the development of proactive behaviors but also increase well-being. This also positively impacts the sustainability of an organization because it is more attractive to employees,

customers, and investors. In other words, when organizations effectively promote advantageous job crafting and avoid costly or dysfunctional job crafting, their employees become more reactive and adaptive to change. This will not only improve workers' sustainable employability but also support sustainable innovation processes of the organizations they work for [28].

**Author Contributions:** Conceptualization, F.S., E.C. and E.I.; Methodology, F.S.; Software, F.S. and E.C.; Validation, F.S.; Formal analysis, F.S. and E.C.; Investigation, F.S., C.G.C., E.D.C. and E.I.; Resources, E.C., E.D.C. and E.I.; Data curation, F.S., E.C. and E.D.C.; Writing—original draft, F.S. and E.I.; Writing—review & editing, F.S., C.G.C., E.D.C. and E.I.; Visualization, E.I.; Supervision, E.C., C.G.C. and E.I.; Project administration, C.G.C. and E.I. All authors have read and agreed to the published version of the manuscript.

**Funding:** This research received no external funding.

**Informed Consent Statement:** Informed consent was obtained from all subjects involved in the study.

**Data Availability Statement:** The data presented in this study are available upon request from the corresponding author. The data are not publicly available due to privacy reasons.

**Conflicts of Interest:** The authors declare no conflict of interest.

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
