# Peer review of "The Active Role of Job Crafting in Promoting Well-Being and Employability: An Empirical Investigation"

_sustainability, doi:10.3390/su16010201_

Round 1

Reviewer 1 Report

Comments and Suggestions for Authors

The paper is well-rounded and adheres to standard research practices. The methodology is sound, and the data analysis using PLS-SEM is appropriate for your study. The results are well presented, and the conclusions drawn are in line with the data.

Currently, the hypotheses are presented towards the end of the introduction. It would be more effective to integrate them throughout the introduction, supporting each hypothesis with relevant literature as it is discussed. This approach would help readers understand the development of your hypotheses in the context of existing research.

The literature review could be expanded to include more studies on job crafting, particularly those that explore its impact on well-being and employability in various contexts. This broader perspective would provide a more comprehensive background for your study and highlight its contribution to the existing body of knowledge. 9.  The Contributions of Population Distribution, Healthcare Resourcing, and Transportation Infrastructure to Spatial Accessibility of Health Care. INQUIRY: The Journal of Health Care Organization, Provision, and Financing,

A Study on the Relationships between Authentic Leadership, Job Crafting, Psychological Capital and Organisational Innovation. International Journal of Management Science and Business Administration,

Diabetes-related avoidable hospitalisations and its relationship with primary healthcare resourcing in China: A cross-sectional study from Sichuan Province. Health & Social Care in the Community, 30(4),

An empirical study of entrepreneurial leadership and fear of COVID-19 impact on psychological wellbeing: A mediating effect of job insecurity

I recommend removing the specific reference to Italy in the title. A title such as "The Active Role of Job Crafting in Promoting Well-Being and Employability: An Empirical Investigation" would be more inclusive and would not limit the perceived relevance of your study to a specific geographic context. This change could broaden the appeal of your findings to an international audience.

Comments on the Quality of English Language

proofread

Author Response

Reviewer 1

Comments and Suggestions for Authors

The paper is well-rounded and adheres to standard research practices. The methodology is sound, and the data analysis using PLS-SEM is appropriate for your study. The results are well presented, and the conclusions drawn are in line with the data.

  1. Currently, the hypotheses are presented towards the end of the introduction. It would be more effective to integrate them throughout the introduction, supporting each hypothesis with relevant literature as it is discussed. This approach would help readers understand the development of your hypotheses in the context of existing research.

Dear Reviewer, thank you for your suggestion, which we appreciate. However, as you have seen from the formulation of the hypotheses, each one contains several relationships between variables, which is why, for the sake of linearity and to avoid confusion, we preferred to put them at the end of the introductory paragraphs. We agree that it would promote linearity if we followed your valuable advice, but it would further break up an already large number of hypotheses, creating a greater sense of disorganization. For this reason, we prefer to leave this first part as it is, ready to modify it if deemed appropriate.

The literature review could be expanded to include more studies on job crafting, particularly those that explore its impact on well-being and employability in various contexts. This broader perspective would provide a more comprehensive background for your study and highlight its contribution to the existing body of knowledge. 

  1. The Contributions of Population Distribution, Healthcare Resourcing, and Transportation Infrastructure to Spatial Accessibility of Health Care. INQUIRY: The Journal of Health Care Organization, Provision, and Financing,
  2. A Study on the Relationships between Authentic Leadership, Job Crafting, Psychological Capital and Organisational Innovation. International Journal of Management Science and Business Administration,
  3. Diabetes-related avoidable hospitalisations and its relationship with primary healthcare resourcing in China: A cross-sectional study from Sichuan Province. Health & Social Care in the Community, 30(4),
  4. An empirical study of entrepreneurial leadership and fear of COVID-19 impact on psychological wellbeing: A mediating effect of job insecurity.

Thank you for the advice. We corroborated the literature with effective and coherent paper.

  1. I recommend removing the specific reference to Italy in the title. A title such as "The Active Role of Job Crafting in Promoting Well-Being and Employability: An Empirical Investigation" would be more inclusive and would not limit the perceived relevance of your study to a specific geographic context. This change could broaden the appeal of your findings to an international audience.

Thank you for the suggestion, we change the title as you proposed.

Reviewer 2 Report

Comments and Suggestions for Authors

It is an original manuscript with an excellent research report with good organization and readability. However, it has a few issues in its construction that need to be solved, as noted below. I encourage authors to improve the manuscript, which presents a potential contribution.

The Introduction section is well justified and theoretically grounded since the authors defined and discussed all concepts used. However, it has a few issues in its construction that need to be solved, as noted below.

a) Update the references, as the vast majority are more than five years old and, as has been pointed out, the world of work and organizations are constantly and rapidly changing.

b) Based on the scientific literature, the study explained and justified the reasons for choosing the variables studied (job insecurity, employability, work engagement, psychological well-being, emotional exhaustion). Why would these be the most appropriate for the study rather than other existing ones?

c) The authors should provide a more detailed description of the sociocultural backgrounds especially regarding gender stereotypes, education and the world of work features, and other cultural aspects. That is necessary to contextualize the discussion of the data. The study may reach mistaken conclusions if this sociocultural contextualization is not provided.

d) Present further secondary data from the context studied to underpin the conceptual model presented.

e) The hypotheses are very well grounded theoretically.

In the method section, the authors provide suitable information regarding the required experiment and its analysis. However, it has a few issues in its construction that need to be solved, as noted below.

a) Briefly describe how the sample specificities (variables) were controlled and discussed in the results.

b) Lack of a more detailed characterization of participants and a brief description of the context studied.

c) Methodological issue: it is necessary to present data indicating the epistemological coherence of the conceptual bases of the instruments used in the research.

The authors presented the report of the results reasonably.

My third concern relates to the discussion section.

a) I appreciated the discussion wherein the authors sought to explore some deeper meaning of their results, although there is little relationship and dialogue between the Introduction and the Results discussion. It is hard to understand how the present research advances knowledge in the topic studied, although it seems it has happened. Thus, the study is more descriptive than explanatory. Include some discussions and analytical hypotheses to make the manuscript more explanatory, improving its quality and relevance.

b) Discuss the impact of the organizational context on the dimensions analyzed in order to avoid conclusions that attribute only the adaptations and changes needed to carry out the work to the employees.

c) The discussion needs to highlight what is specific to the context analyzed and what can be generalized in terms of the subject studied to foster international interest in the manuscript.

d) Due to the journal scope, sustainability should be included as one of the axes of discussion.

Author Response

Reviewer 2

Comments and Suggestions for Authors

It is an original manuscript with an excellent research report with good organization and readability. However, it has a few issues in its construction that need to be solved, as noted below. I encourage authors to improve the manuscript, which presents a potential contribution.

The Introduction section is well justified and theoretically grounded since the authors defined and discussed all concepts used. However, it has a few issues in its construction that need to be solved, as noted below.

  1. Update the references, as the vast majority are more than five years old and, as has been pointed out, the world of work and organizations are constantly and rapidly changing.

Thank you for the suggestion. We update the references as you requested. All are from 2000 onwards except one, but most now refer to the last five years.

  1. Based on the scientific literature, the study explained and justified the reasons for choosing the variables studied (job insecurity, employability, work engagement, psychological well-being, emotional exhaustion). Why would these be the most appropriate for the study rather than other existing ones?

Thank you for pointing out these aspects. As explained in the paper and reported by you, there are many variables that measure wellbeing in employees, also and especially from a sustainable perspective. Since we could not carry out a study including all variables, we decided to measure how a construct related to proactivity, and therefore part of personal sustainability, can activate behaviors that, according to the JD-R model, are categorized into positive and negative outcomes. In addition, we wanted to investigate whether such behavior is dependent on conditions made more frequent by the pandemic, such as job insecurity. We have, however, included supporting sentences.

  1. The authors should provide a more detailed description of the sociocultural backgrounds especially regarding gender stereotypes, education and the world of work features, and other cultural aspects. That is necessary to contextualize the discussion of the data. The study may reach mistaken conclusions if this sociocultural contextualization is not provided.

Thanks for the note. We included a discussion about the influence of socio-cultural factors and caution in generalizing results in the subsection "Limits".

  1. Present further secondary data from the context studied to underpin the conceptual model presented.

Dear reviewer, we are not sure if we have understood your comment correctly. In any case, in section 3.4 we included descriptive lines concerning literature sources that agree in defining the conceptual model adopted (the JD-R) as appropriate in the study of organizational phenomena belonging to different fields (school, university, disability, companies).

The hypotheses are very well grounded theoretically.

In the method section, the authors provide suitable information regarding the required experiment and its analysis. However, it has a few issues in its construction that need to be solved, as noted below.

  1. Briefly describe how the sample specificities (variables) were controlled and discussed in the results.
  2. Lack of a more detailed characterization of participants and a brief description of the context studied.

Dear reviewer, thank you for once again reminding us to put the results into context. We included several lines on this, focusing on sample characteristics and related considerations in the limits section, as these are appropriate aspects to consider when generalizing the results. Moreover, to highlight the principal characteristics of the studied context, we added Table 2.

A more detailed description of the context of the study is also not possible, as the 641 workers involved (by snowball sampling) belonged to different, heterogeneous and hardly comparable contexts.

  1. Methodological issue: it is necessary to present data indicating the epistemological coherence of the conceptual bases of the instruments used in the research.

Thank you. We added at section 3.4 references through which epistemological coherence of the adopted tools was confirmed.

The authors presented the report of the results reasonably.

My third concern relates to the discussion section.

  1. I appreciated the discussion wherein the authors sought to explore some deeper meaning of their results, although there is little relationship and dialogue between the Introduction and the Results discussion. It is hard to understand how the present research advances knowledge in the topic studied, although it seems it has happened. Thus, the study is more descriptive than explanatory. Include some discussions and analytical hypotheses to make the manuscript more explanatory, improving its quality and relevance.

Thanks for the note. We sought to logically link introduction and discussion, distancing ourselves from a mere description of the results. In the lines, there is an attempt to interpret what was found to provide practical indications also in terms of sustainability.

  1. Discuss the impact of the organizational context on the dimensions analyzed in order to avoid conclusions that attribute only the adaptations and changes needed to carry out the work to the employees.

We added some lines on the role of organizational context.

  1. The discussion needs to highlight what is specific to the context analyzed and what can be generalized in terms of the subject studied to foster international interest in the manuscript.

Dear Reviewer, we discussed these two aspects in Limitations and Recommendation Section.

  1. Due to the journal scope, sustainability should be included as one of the axes of discussion.

We included sustainability as a driver for discussion.

Reviewer 3 Report

Comments and Suggestions for Authors

Dear editor,

Dear authors,

Thank you for inviting me to review this paper. the objective of this study is to assess the strategic role of job crafting in the relationship between job insecurity and work engagement, psychological wellbeing, and emotional exhaustion, and also to specifically investigate how much age impacts on these behaviors and on the relationship between job crafting and employability.

This paper has significant contributions and can be accepted after several revisions below:

The abstract is well written.

Introduction: the authors not provide contribution of their study, novelty research and the explanation of the structure of paper in the last paragraph in introduction section

Line 74. Why author just analysis the role of age, gender and work crafting experiences will also potential to affect their behavior.

Methodology section: figure 2 need to revise to table format.

Why author use SEM approach? what is SEM? Please explain about CB-SEM? Compare with PLS-SEM and explain more. Added citations.

Line 340-353. Added the table for explain respondent clearer. Table easier to understand for reader that descriptive paragraph.

Data collection and data analysis is missing. In this section you need to explain how you get the data, consent form and how the step to analysis the data.

According to T. T. Wijaya, P. Jiang, and M. Mailizar, “Predicting Factors Influencing Preservice Teachers ’ Behavior Intention in the Implementation of STEM Education Using Partial Least Squares Approach,” Sustain., 2022. In structural model evaluation, an analysis of the F2 value alsoneeds to be performed. This construct explains the effect of exogenous on endogenous variables to determine changes in the R2 value when specific exogenous determinants are excluded from the model. besides, Moderating Effect Analysis of Gender and Age also can be analysis. This paper may improve the quality of this paper.

The conclusion section needs to separate.

The limitations sections is missing

Recommendations section and implications section is missing.

References section can be improved.

Author Response

Reviewer 3

Comments and Suggestions for Authors

Dear authors,

Thank you for inviting me to review this paper. the objective of this study is to assess the strategic role of job crafting in the relationship between job insecurity and work engagement, psychological wellbeing, and emotional exhaustion, and also to specifically investigate how much age impacts on these behaviors and on the relationship between job crafting and employability.

This paper has significant contributions and can be accepted after several revisions below:

The abstract is well written.

  1. Introduction: the authors not provide contribution of their study, novelty research and the explanation of the structure of paper in the last paragraph in introduction section.

Dear reviewer, thank you for your suggestion. We added some lines relative to the novelty of our study.

  1. Line 74. Why author just analysis the role of age, gender and work crafting experiences will also potential to affect their behavior.

Thank you for the note. We chose to focus only on age as it is a factor strongly connected with the concept of sustainability and employability. Moreover, the role of age on job crafting behavior was studied and proven in other papers, taken as reference (Ingusci, 2018; Ingusci et al., 2019), so the assumption was to corroborate these assumptions. What you say however is true, thus we decided to include it in the limitations and future perspectives.

  1. Methodology section: figure 2 need to revise to table format.

We transformed Figure 2 in Table 1.

  1. Why author use SEM approach? what is SEM? Please explain about CB-SEM? Compare with PLS-SEM and explain more. Added citations.

Thanks for the note. Although the methods and results section already went into this in great detail, we have created a subsection called 3.1.            Latent variables approach: PLS-SEM and CB-SEM, in which we explained at the beginning what the latent variables method consists of. We have also included further citations.

  1. Line 340-353. Added the table for explain respondent clearer. Table easier to understand for reader that descriptive paragraph.

Dear reviewer, I assume you are referring to the lines referred to in the sample description, not the ones quoted in the document. However, we have added a summary table to make everything clearer.

  1. Data collection and data analysis is missing. In this section you need to explain how you get the data, consent form and how the step to analysis the data.

We added the subparagraph called “Data collection and data analysis” to deepen the aspect you requested.

  1. According to T. T. Wijaya, P. Jiang, and M. Mailizar, “Predicting Factors Influencing Preservice Teachers ’ Behavior Intention in the Implementation of STEM Education Using Partial Least Squares Approach,” Sustain., 2022. In structural model evaluation, an analysis of the F2 value also needs to be performed. This construct explains the effect of exogenous on endogenous variables to determine changes in the R2 value when specific exogenous determinants are excluded from the model. besides, Moderating Effect Analysis of Gender and Age also can be analysis. This paper may improve the quality of this paper.

Thank you for the suggestion. We added the power of effect sizes in Table 7 and performed a frequency analysis on their strength. Regarding the moderation analysis of gender and age, the purpose of the paper was to substantiate the assumptions regarding the relationship between age and job crafting of Ingusci, 2018 and Ingusci et al., 2019. The relationship, therefore, is based on a theoretical assumption. Therefore, we did not consider elaborating on the moderating relationship of age and gender, although it may give the article more prominence. It might be an idea for future studies.

  1. The conclusion section needs to separate.

Dear Reviewer, as you can see in the attached paper, the conclusion section is already separate from the discussion.

  1. The limitations sections is missing

Thank you for the advice. We added limitation section.

  1. Recommendations section and implications section is missing.

We added recommendations and implications section.

  1. References section can be improved.

Thank you for the suggestion. We update the references as you requested. All are from 2000 onwards except one, but most now refer to the last five years.

Round 2

Reviewer 2 Report

Comments and Suggestions for Authors

It is an original manuscript with an interesting research report with very good organization and readability. The authors did a very good job in reviewing the manuscript. I recommend it for publication.

Reviewer 3 Report

Comments and Suggestions for Authors

Dear editor, I have read the revision version of manuscript. now manuscript ready to publish.